# The Spanish Version of the International Index of Erectile Function: Adaptation and Validation

**DOI:** 10.3390/ijerph20031830

**Published:** 2023-01-19

**Authors:** Esther Díaz-Mohedo, Antonio Meldaña Sánchez, Francisco Cabello Santamaría, Elena Molina García, Sofía Hernández Hernández, Fidel Hita-Contreras

**Affiliations:** 1Department of Physiotherapy, Faculty of Health Science, Ampliación de Campus de Teatinos, University of Malaga, 29071 Málaga, Spain; 2PELVICUS, Pelvic Floor Service, 28500 Madrid, Spain; 3Andalusian Institute of Sexology and Psychology, 29001 Málaga, Spain; 4Freelance Physiotherapist, 29720 Málaga, Spain; 5Department of Physiotherapy, Faculty of Health Science, University of Granada, 18071 Granada, Spain; 6Department of Physiotherapy, Faculty of Health Science, University of Jaén, 23071 Jaén, Spain

**Keywords:** erectile function, Spanish, validation, psychometric

## Abstract

Background: The International Index of Erectile Function (IIEF) is a widely employed questionnaire in urology to assess erectile dysfunction (ED) in both clinical research and practice. Objective: To translate and culturally adapt the Spanish version of the International Index of Erectile Function (IIEF) and to analyze its psychometric properties in Spanish men with erectile dysfunction (ED). Methods: Firstly, direct and reverse translations were performed. Secondly, a pilot study was carried out on 23 patients with the lowest possible education level without being illiterate. Finally, 170 participants completed the IIEF. Test–retest reliability, internal consistency and construct validity (exploratory factor analysis) were assessed. Concurrent and divergent validity were evaluated with the Hospital Anxiety and Depression Scale (HADS) and the 12-item Short-Form Health Survey (SF-12), respectively. Discriminant validity (with and without anxiety or depression) was calculated using a receiver-operating characteristic curve analysis. Results: High internal consistency (Cronbach’s alpha = 0.968, total score) and moderate-to-excellent test–retest reliability were found. The factor analysis showed a two-factor structure (explained variance of 77.34%). Significant correlations of the IIEF total score (*p* < 0.01) and domains (*p* < 0.05) with HADS anxiety and depression scores were observed (concurrent validity), while non-significant correlations with SF-12 physical and mental summary scores were found (divergent validity). The IIEF total score could discriminate between participants with and without anxiety (*p* < 0.05) and depression (*p* < 0.01), with an optimal cut-off point of <39.50 for both anxiety (48.30% sensitivity and 78.75% specificity) and depression (50.00% sensitivity and 81.01% specificity). Clinical implications: The psychometric properties of the IIEF have not been analyzed in Spanish people to date. Strengths and Limitations: The Spanish version of the IIEF was shown to be capable of discriminating between men with erectile dysfunction with and without depression or anxiety. There are some limitations to this study that should be noted. It was conducted on Spanish participants, and, thus, it should be employed with caution in other Spanish-speaking countries. This study was carried out on a selected population, and, therefore, the generalizability of its results to other populations might be limited. Moreover, a large majority of the participants (89.41%) had secondary or higher education. Future studies should be performed on a more general population with a varied geographical and educational background. Conclusions: The Spanish IIEF is a valid and reliable instrument for assessing erectile function among Spanish men with ED.

## 1. Introduction

Erectile dysfunction (ED) is defined as the inability to achieve or maintain a rigid penile erection suitable for satisfactory sexual intercourse [1]. It is one of the most frequent sexual dysfunctions worldwide [2], and it has been described as an important cause of decreased quality of life people with the condition [3]. ED is related to several psychosocial problems such as anxiety, depression, frustration, decreased self-esteem and confidence, and limited intimacy [4]. Therefore, ED has a great quality-of-life burden in both people with the condition and their partners, as well as a significant economic impact [5].

Several studies have attempted to determine the prevalence of ED. For instance, Derogatis and Burnett [6] reported that the estimated overall prevalence ranges from 10% to 20% worldwide, taking into account differences in methodology, definitions and study population. In a study conducted on 2476 non-institutionalized Spanish men (25–75 years), the prevalence of ED varied from 12.1% to 18.9% according to the method used [7]. By contrast, Feldman et al. [8], in the Massachusetts Men’s Aging Study, described a prevalence of 52% in men aged 40–70 years.

ED can be diagnosed using several methods, such as nocturnal penile monitoring, imaging techniques, neurological studies and psychological assessment [9,10]. Validated patient self-report techniques have been shown to be useful in evaluating ED, as well as in monitoring treatment response [11].

The International Index of Erectile Function (IIEF) is a widely employed questionnaire in urology to assess ED in both clinical research and practice [12]. The IIEF was first developed by Rosen et al. in 1996 [13], and it consists of fifteen items, whose scores range from zero (or one) to five. The items are grouped into five domains: erectile function (six items), orgasmic function (two items), sexual desire (two items), intercourse satisfaction (three items) and overall satisfaction (two items). Higher scores indicate better erectile function.

The IIEF has been linguistically validated and is currently available in several languages worldwide. The psychometric analysis of the Spanish version has been carried out on Peruvian and Chilean populations [14,15], and has concluded that the IIEF is a valid and reliable instrument for the study of ED in clinical and research settings; however, to the best of our knowledge, the psychometric properties of the IIEF have not been analyzed among Spanish people to date.

The objective of the present study was to translate and culturally adapt the Spanish version of the IIEF and to analyze the psychometric properties of the Spanish version of the IIEF among Spanish men with erectile dysfunction by evaluating internal consistency and test–retest reliability, as well as the construct, concurrent and divergent validity.

## 2. Methods

### 2.1. Participants

The present study took place from February to November 2020. Initially, 230 men with erectile dysfunction who attended the Andalusian Institute of Sexology and Psychology in Málaga (Spain) were contacted, of whom 170 finally participated in this study. According to Kline [16], the sample size of this study was appropriate (being at least 100 participants). This study was authorized by the Ethics Committee of the University of Málaga (Málaga, Spain; Registration No.: 649 CEUMA: 120-2020-H), and it was conducted in line with the Declaration of Helsinki, good clinical practices, and all applicable laws and regulations. Prior to the study’s inception, all participants provided their written informed consent. The inclusion criteria were older adults diagnosed with ED who were capable of answering the questionnaires and who had engaged in at least one sexual activity over the last four weeks. People were excluded if they were illiterates or diagnosed with malignant neoplasia, psychiatric disorder, drug or alcohol abuse and local anatomical malformations that affect sexual function.

Descriptive data, such as age, education level, smoking and drinking habits, sport practice and diagnosed diseases, were collected before the questionnaires were administered.

### 2.2. Questionnaires

Translation and cross-cultural adaptation were conducted according to guidelines recommended by the International Quality of Life Assessment project for cross-cultural translation [17]. Firstly, the questionnaire was translated into Spanish by two independent bilingual speakers, who were both unaware of the purpose of the translation and of the fact that another translator was doing the same task. The Spanish translations were then compared for inconsistencies. The two Spanish versions were then translated back into English, likewise blindly and independently, by two different bilingual speakers. Each English translation was then compared with the original English IIEF and checked for inconsistencies. The Spanish version was then jointly reviewed by a bilingual team comprising the four translators, three pelvic floor specialists and two methodologists to assess the necessity of performing a cultural adaptation and to fine-tune it for use among Spanish patients. They again compared the Spanish version with the original English version to detect errors of interpretation and nuances that might have been missed. This version was finalized after small changes were made by consensus. The time required to complete the questionnaire was 4–5 min.

In the second phase, a pilot study was carried out on 23 patients with the lowest educational level possible without being illiterate. Each patient was given the IIEF and asked by the clinician about their comprehension of the meaning of each of the questions.

In order to assess the test–retest reliability, the 170 participants were asked to recomplete the Spanish version of the IIEF questionnaire 21 days later. This period was long enough for the participants to forget their previous answers, but short enough to ensure that their erectile function outcomes did not change.

The Hospital Anxiety and Depression Scale (HADS) is a well-known self-report measure of both anxiety and depression [18,19]. This questionnaire consists of 14 items, 7 concerning anxiety and 7 concerning depression. Scores for both anxiety and depression subscales range from 0 to 21, where higher scores indicate more severe symptoms. A threshold of ≥11 indicates probable presence of anxiety or depressive disorders.

The 12-item Medical Outcomes Study Short Form Health Survey (SF-12), version 2.0, is a generic questionnaire used to evaluate health-related quality of life (HRQoL) [20]. It consists of 12 items, which provide information about eight domains: self-perceived general health, bodily pain, physical functioning, physical role, vitality, social functioning, mental health and emotional role. The SF-12 also provides two separate main scores: physical and mental component summary scores (PCS and MCS, respectively). Higher scores indicate better general HRQoL.

### 2.3. Statistical Analysis

Mean and standard deviation (SD) were used for the continuous variables, while frequencies and percentages were employed for the categorical variables. The internal consistency of the instrument was determined using Cronbach’s α coefficient. A Cronbach’s α coefficient > 0.70 was considered acceptable [21]. The test–retest reliability was assessed using Intraclass Correlation Coefficients (ICC_2,1_) by Shrout and Fleiss. According to the ICC, test–retest reliability was classified as poor (< 0.40), moderate (0.40–0.75), substantial (0.75–0.9) or excellent (>0.90) [22]. Floor and ceiling effects were assessed by determining the number of participants with the minimum (0) or maximum (100) score. These effects were present if ≥15% of the participants obtained the minimum or maximum score. An exploratory factor analysis (principal component analysis) was performed to assess construct validity. A varimax rotation of factors was employed and the Kaiser–Meyer–Olkin (KMO) was calculated to measure the sampling adequacy. Concurrent validity (HADS anxiety and depression scores) and divergent validity (SF-12 PCS and MCS scores) were analyzed using Pearson’s correlation coefficient. Discriminant validity was studied by comparing the mean IIEF total scores of the subjects with and without anxiety and depression (Student’s *t*-test). A receiver-operating characteristic (ROC) curve analysis was employed to assess the accuracy of the Spanish version of the IIEF total score in discriminating between participants with/without anxiety and with/without depression, and a cut-off point was obtained for each [23]. We also calculated the area under the ROC curve (AUC) as a measure of how well the IIEF was able to distinguish between groups with/without anxiety and depression. AUC was considered statistically significant when the 95% confidence interval did not include the 0.5 value. The level of statistical significance was set at *p* < 0.05. All the analyses were performed using the SPSS 20.0 statistical package (SPSS Inc., Chicago, IL, USA).

## 3. Results

A total of 170 men completed all the questionnaires in this study. Table 1 displays the descriptive characteristics. The mean age of the participants was 60.53 ± 9.26 years. Most of the participants were nonsmokers (65.88%) and had secondary or higher education (89.41%). The mean total score of the IIEF was 33.99 ± 16.34.

Regarding the analysis of internal consistency, the Cronbach’s alpha value of the total score of the Spanish IIEF was 0.968, which indicates a high level of internal consistency. This was also observed for every domain, where Cronbach’s alpha values were erectile function (0.955), intercourse satisfaction (0.891), orgasmic function (0.938), sexual desire (0.923) and overall satisfaction (0.912). When any of the items were deleted, the Cronbach’s alpha value ranged between 0.964 and 0.968 without items 6 or 12. When analyzing item-to-item comparison, a significant positive correlation was observed (all *p* < 0.001), with values ranging from 0.414 (items 6 and 13) to 0.892 (items 9 and 10). The item-to-total score values ranged from 0.709 (item 6) to 0.918 (item 8).

As for the test–retest reliability (Table 2), three participants did not complete the questionnaire after the three-week period. The analysis showed moderate to substantial correlations for all the items of the Spanish version of the IIEF except for item 5 (excellent). The total score reliability was excellent (ICC: 0.986. 95% CI: 0.982–0.990).

No floor and ceiling effects were observed for the Spanish IIEF total score, given that less than 15% of the participants scored either the minimum (5 points; *n* = 12) or the maximum (75 points; *n* = 0) score.

The exploratory factor analysis of the Spanish IIEF (using the principal components option) suggested a two-factor structure (Table 3). The items corresponding to the original IIEF domains “erectile function” (1–5 and 15) and “intercourse satisfaction” (6–8) loaded in the first factor, together with the item “ejaculation frequency” (of the original domain “orgasmic function”). By contrast, the second factor contained the items of the original IIEF “sexual desire” and “overall satisfaction”, as well as the item “orgasm frequency” (originally “orgasmic function”). The total variance explained by this model was 77.34%, with sampling adequacy measured as KMO = 0.938 (*p* < 0.001). Therefore, the sample can be considered adequate for the purposes of this analysis.

In the concurrent validity assessment (Table 4), our results indicate that the two domains and the total score of the Spanish IIEF showed statistically significant negative correlations with both the HADS anxiety and depression scores. As for divergent validity, the SF-12 was used, and non-significant low correlations were found between all the domains and the total score of the Spanish version of the IIEF with the SF-12 PCS and MCS.

Finally, in order to assess the discriminant validity, the IIEF total score was compared between participants with and without anxiety and depression, according to the HADS scores. We found statistically significant lower IIEF total scores among participants with anxiety (*n* = 80; *p* = 0.024) and depression (*n* = 79; *p* = 0.007). In the ROC curve analysis (Figure 1), the IIEF total score demonstrated significant capacity to discriminate between participants with and without anxiety and depression. Our results show a statistically significant AUC regarding the presence of anxiety (0.635; *p* = 0.003) and depression (0.649; *p* = 0.001), with a cut-off point (IIEF total score) of <39.50 with 48.30% sensitivity and 78.75% specificity for detecting anxiety, and a cut-off point of <39.50 with 50.00% sensitivity and 81.01% specificity for detecting depression.

## 4. Discussion

The goal of this study was to assess the psychometric properties of the Spanish version of the IIEF for Spanish patients with erectile dysfunction. Our results showed that the Spanish IIEF is a valid and reliable instrument for evaluating erectile function in this population and that it is able to discriminate between participants with and without anxiety and depressive disorders.

When analyzing the internal consistency, Rossen et al. [13] found that the IIEF total score as well as the erectile and orgasmic function domains were highly consistent (with Cronbach’s alpha values higher than 0.90), with a satisfactory degree of consistency (values > 0.70) for the other domains. Hernández et al. [15] and Wiltink et al. [24] also obtained a high internal consistency for the total score of the Chilean and German IIEF versions (Cronbach’s alpha values of 0.97 and 0.95, respectively). The findings of the present study are in accordance with those previously described, with the Spanish IIEF showing alpha values greater than 0.90 for the total score and the two domains obtained in the factorial analysis.

To evaluate the test–retest reliability, all the participants recompleted the questionnaire three weeks later. The results showed moderate–substantial ICC values for all the Spanish IIEF items; however, for the “maintenance ability” item and the total score, the correlation was excellent. These findings are in line with those previously described in the development and validation analysis of the IIEF, as well as in other validations, such as the Iranian [25] and Brazilian versions [26]. By contrast, Zegarra et al. [14], in the Peruvian validation, found a high Spearman’s correlation coefficient for the total score, but these correlations ranged from weak to moderate for the domains. The present analysis also showed that there were no floor or ceiling effects, which means that <15% of the participants answered with the highest or lowest score for the items and total score.

As for the construct validity analysis, our exploratory factor analysis yielded a two-factor structure, with one larger factor that seemed to contain the items related to a physical aspect of erectile function, and a smaller one that referred to a psychological aspect. The first factor comprised the items corresponding to the original erectile function and intercourse satisfaction domains, whereas sexual desire and overall satisfaction were included in the second factor. As for the orgasmic function domain, the item “orgasm frequency” also loaded in this last smaller factor, while the other one (ejaculation frequency) did so in the first factor. In the German validation, Wiltink et al. [24] also found a two-factor structure, where twelve items loaded in one factor, which the authors named “sexual function” (items 1–11 and 15), and the remaining three items (12–14) loaded in another factor, called “sexual desire”. Similarly, Quinta Gomes and Nobre [27], in the Portuguese validation, also reported two factors, with one encompassing sexual desire and overall satisfaction, as well as intercourse satisfaction, which explained approximately 55% of the total variance. The results of our exploratory factor analysis reveal that the total variance explained by this two-factor model was 77.34%, with KMO = 0.938, that is, greater than 0.60, which is the minimum value identified as adequate for the analysis [28]. The total variance explained in the German IIEF validation was 70% [24].

Divergent validity can be explained as a lack of association with scales that do not directly assess erectile function. Therefore, we employed the SF-12 questionnaire. The Spanish IIEF total score and factors showed very low or non-significant correlations with the SF-12 MCS and PCS scores, demonstrating the divergent validity of the Spanish IIEF. Pakpour et al. found an appropriate divergent validation of the Iranian version of the IIEF when compared with the Life Satisfaction Questionnaire [25]. Both depression and anxiety, though, are associated with ED, and the nature of these associations is bidirectional. It has been estimated that 25% of men who experience depressive symptoms have ED [29]. A recent study described that 79.82% and 79.56% of Chinese ED patients had anxiety and depression, respectively [30]. In this study, the HADS was used to assess concurrent validity, and our results determined that lower values (worse erectile function) of the Spanish IIEF total score, and the two domains obtained in the exploratory factor analysis were significantly correlated with both higher HADS anxiety and depression scores (greater symptoms). Finally, as for the discriminant validity, we evaluated the ability of the Spanish IIEF total score to differentiate between people with and without clinical anxiety and depression, as assessed with the HADS. Our results show that participants with clinical anxiety and depression had significantly lower scores in the IIEF total score. Moreover, values lower than 39.50 could discriminate between participants with anxiety (48.30% sensitivity and 78.75% specificity) and depression (50.00% sensitivity and 81.01% specificity).

There are some limitations to this study that should be noted. It was conducted on Spanish participants, and, thus, it should be employed with caution in other Spanish-speaking countries. This study was carried out on a selected population, and, therefore, the generalizability of its results to other populations might be limited. Moreover, a large majority of the participants (89.41%) had secondary or higher education. A confirmatory factor analysis was not performed. Future studies should be carried out on a more general population, comprising people from different geographical locations and educational backgrounds, using a confirmatory factor analysis.

## 5. Conclusions

From the findings of the present study, it can be concluded that in men with erectile dysfunction, the Spanish IIEF has good internal consistency and test–retest reliability, with no ceiling or floor effects. It also has satisfactory construct, concurrent and divergent validity. In addition, the Spanish version of the IIEF was shown to be capable of discriminating between men with erectile dysfunction with and without depression or anxiety.

## Figures and Tables

**Figure 1 ijerph-20-01830-f001:**
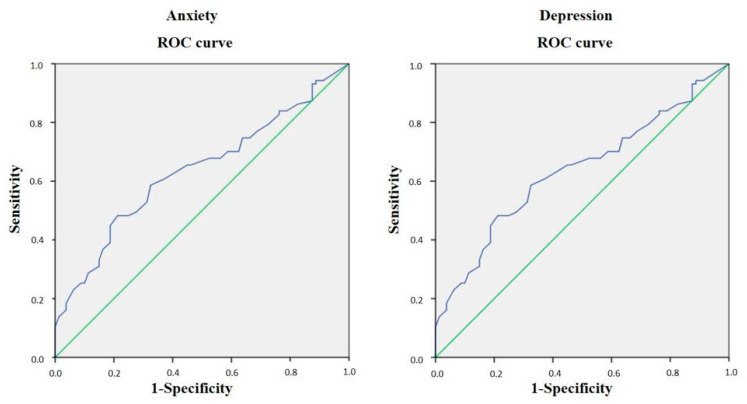
The ROC curve of the Spanish IIEF total score for discriminating among participants with and without anxiety and depression. ROC: receiver-operating characteristic.

**Table 1 ijerph-20-01830-t001:** Descriptive data of the sample (*n* = 170).

Participants’ Characteristics						
Age		60.53 ± 9.26	Diagnosis	DM	No	134 (78.82)
Education	No studies	1 (0.59)			Yes	36 (21.18)
	Primary	17 (10.00)		CVD	No	102 (60.00)
	Secondary	61 (35.88)			Yes	66 (38.82)
	Higher	91 (53.53)		MetS	No	160 (94.12)
Beer intake (per day)	Do not drink	72 (42.35)			Yes	10 (5.88)
	1–2	63 (37.06)		Hypogonadism	No	166 (97.65)
	3–4	28 (16.47)			Yes	4 (2.35)
	5–6	6 (3.53)		Urological diseases	No	155 (91.18)
	>6	1 (0.59)			Yes	15 (8.82)
Alcohol intake (per day)	Do not drink	108 (63.53)		Prostate cancer	No	151 (88.82)
	1–2	42 (24.71)			Yes	19 (11.18)
	3–4	20 (11.76)	IIEF Total score			33.99 ± 16.34
Smoker	No	112 (65.88)	HADS Anxiety			10.44 ± 4.68
	Yes	58 (34.12)	HADS Depression			10.09 ± 4.83
Sport practice	No	78 (45.88)	SF-12 PCS			59.75 ± 18.53
	Yes	92 (54.12)	SF-12 MCS			57.30 ± 14.48

Values expressed as mean ± standard deviation and frequency (percentage). IIEF: International Index of Erectile Function. SD: standard deviation. DM: diabetes mellitus. CVD: cardiovascular disease. MetS: metabolic syndrome. HADS: Hospital Anxiety and Depression Scale. SF-12 MCS: Short-Form Health Survey 12-item Mental Component Summary. SF-12 PCS: Short-Form Health Survey 12-item Physical Component Summary.

**Table 2 ijerph-20-01830-t002:** Test–retest reliability of the Spanish version of the IIEF.

	Test–Retest Reliability (*n* = 167)			
	ICC	95% CI		*p*-Value
1. Erection frequency	0.634 **	0.534	0.717	<0.001
2. Erection firmness	0.633 **	0.533	0.716	<0.001
3. Penetration ability	0.826 *	0.772	0.869	<0.001
4. Maintenance frequency	0.745 **	0.655	0.812	<0.001
5. Maintenance ability	0.975 ***	0.966	0.981	<0.001
6. Intercourse frequency	0.568 **	0.335	0.713	<0.001
7. Intercourse satisfaction	0.747 **	0.671	0.807	<0.001
8. Intercourse enjoyment	0.816 *	0.757	0.862	<0.001
9. Ejaculation frequency	0.487 **	0.281	0.634	<0.001
10. Orgasm frequency	0.755 *	0.671	0.818	<0.001
11. Desire frequency	0.857 *	0.810	0.893	<0.001
12. Desire level	0.542 **	0.420	0.644	<0.001
13. Overall satisfaction	0.513 **	0.393	0.617	<0.001
14. Relationship satisfaction	0.443 **	0.287	0.571	<0.001
IIEF item 15	0.513 **	0.389	0.618	<0.001
IIEF total score	0.986 ***	0.982	0.990	<0.001

IIEF: International Index of Erectile Function. ICC: Intraclass Correlation Coefficient (* substantial, ** moderate, *** excellent); CI: confidence interval.

**Table 3 ijerph-20-01830-t003:** Exploratory factor analysis of the Spanish version of the IIEF.

Exploratory Factor Analysis (*n* = 170)		
Original IIEF Domains	Factor 1	Factor 2
Erectile Function	0.800	
Erectile Function	0.860	
Erectile Function	0.840	
Erectile Function	0.841	
Erectile Function	0.785	
Intercourse Satisfaction	0.749	
Intercourse Satisfaction	0.752	
Intercourse Satisfaction	0.762	
Orgasmic Function	0.622	
Orgasmic Function		0.689
Sexual Desire		0.887
Sexual Desire		0.856
Overall Satisfaction		0.638
Overall Satisfaction		0.686
Erectile Function	0.663	

IIEF: International Index of Erectile Function.

**Table 4 ijerph-20-01830-t004:** Concurrent and divergent validity of the Spanish version of the IIEF total score and domains (*n* = 150).

		HADS Anxiety		HADS Depression	
	Spanish IIEF	r	*p*-value	r	*p*-value
Concurrent validity	Domain 1	−0.188	0.014	−0.181	0.018
	Domain 2	−0.266	<0.001	−0.276	<0.001
	Total score	−0.214	0.005	−0.214	0.005
		SF-12 PCS		SF-12 MCS	
	Spanish IIEF	r	*p*-value	r	*p*-value
Divergent validity	Domain 1	0.011	0.892	0.048	0.537
	Domain 2	0.030	0.697	0.026	0.735
	Total score	0.009	0.906	0.036	0.646

IIEF: International Index of Erectile Function. HADS: Hospital Anxiety and Depression Scale. r: Pearson’s correlation coefficient. VAS: Visual Analogue Scale. SF-12 MCS: Short-Form Health Survey 12-item Mental Component Summary. SF-12 PCS: Short-Form Health Survey 12-item Physical Component Summary.

## Data Availability

Data sharing not applicable.

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
