# Peer review of "The Spanish Version of the International Index of Erectile Function: Adaptation and Validation"

_ijerph, 2023, doi:10.3390/ijerph20031830_

Round 1

Reviewer 1 Report

Linguistic and psychometric validation of a questionnaire or test is a useful tool for both research and clinical assessment. Here the authors presented the Spanish validation of the IIEF an instrument for the assessment of erectile dysfunction widely used by professionals in the field. The article is well written and the methodology is clear and appropriate. I only have a couple of suggestions that might be useful to improve the readability and understanding of the results.
In particular, I think it might be useful to report the EFA in a separate table, perhaps comparing the indices obtained by the authors on the analyzed sample with those of the original validation. Second, in light of the data obtained, I think it might be interesting to compare the results with similar populations, such as those in southern Europe, if available, with culturally and socially similar samples (e.g., Portugal, Italy, and Greece).
Finally, since they are mentioned in the discussion, I would introduce the results of the various validations already in the introduction. I think that the in the future the results can be expanded with a CFA and a study on a sample that has not been diagnosed to actually test its diagnostic validity.

Reviewer 2 Report

Dear editors,

Have a nice day. 

This paper analyzes the psychometric properties of the Spanish version of the IIEF in Spanish men with erectile dysfunction by evaluating internal consistency and test-retest reliability, as well as the construct, concurrent, and divergent validity. Though some related work has appeared in the literature, this work chooses a different sample--Spanish men--which is the unique aspect of this work. I found it a nicely drafted work where: 

Introduction: provides the research gaps and contribution of this work. However, Lines 68-69 about the research gap could possibly offer the evidence taken from studies- Ref. 14 and 15.      

Methods: are clearly stated and technically fine. The sample of 230 men is representative. 

Results: are clearly stated and graphed. 

Conclusions: are in line with the questions raised in the introductory part. 
